# Evaluation of Vaccine Hesitancy and Anxiety Levels among Hospital Cleaning Staff and Caregivers during COVID-19 Pandemic

**DOI:** 10.3390/vaccines10091426

**Published:** 2022-08-30

**Authors:** Sami Akbulut, Ayse Gokce, Gulseda Boz, Hasan Saritas, Selver Unsal, Ali Ozer, Mehmet Serdar Akbulut, Cemil Colak

**Affiliations:** 1Department of Surgery and Liver Transplant Institute, Inonu University Faculty of Medicine, Malatya 44280, Turkey; 2Department of Public Health, Inonu University Faculty of Medicine, Malatya 44280, Turkey; 3Department of Biostatistics, Bioinformatics and Medical Informatics, Inonu University Faculty of Medicine, Malatya 44280, Turkey; 4Department of Surgical Nursing, Siirt University Faculty of Health Sciences, Siirt 56100, Turkey; 5Department of Nursing Service, Inonu University Faculty of Medicine, Malatya 44280, Turkey; 6Department of Social Work, Bingol State Hospital, Bingol 12000, Turkey

**Keywords:** COVID-19, healthcare professionals, vaccine hesitancy, anxiety

## Abstract

It is important to vaccinate individuals working in the field of health who are more at risk compared to society during the pandemic period. The aim of this study was to evaluate the vaccine hesitancy and anxiety levels of hospital cleaning staff and caregivers during the COVID-19 pandemic. This descriptive type cross-sectional study was conducted with 460 hospital cleaning staff and caregivers. Demographic and social characteristics form, Coronavirus Anxiety Scale (CAS), and Vaccine Hesitancy Scale (VHS) adapted to the pandemic were used in the questionnaire form used to collect the data of the study. It was determined that the rates of hesitation against the COVID-19 vaccine and childhood vaccine were 42.2% (*n* = 194) and 10.9% (*n* = 50), respectively. Less than half of the participants (44.6%) believe that the COVID-19 vaccine is protective. COVID-19 anxiety (CAS score ≥ 9 point) was detected in 19.6% of participants and statistically significant differences were found between patients with (*n* = 90) and without (*n* = 370) anxiety regarding gender (*p* < 0.001), working unit (*p* = 0.002), vaccination status (*p* = 0.023) and history of psychological disease (*p* = 0.023). It has been shown that the VHS-total scores of those who are not vaccinated, those who are hesitant about vaccination, those who do not think that the vaccine is protective, and those who state that there is no need for a legal obligation in vaccination are higher. When participants were asked about the most anxious situation during the COVID-19 period, the highest response rate was 62.4% for my parents’ exposure to COVID-19. The most anxious situation among participants is their parents’ exposure to COVID-19. Although participants are highly vaccinated, they have serious hesitancy about the COVID-19 vaccine. This study also showed that there was a parallel relationship between COVID-19 anxiety and vaccine hesitancy.

## 1. Introduction

In December 2019, bizarre and severe pneumonia was observed in Wuhan City of Hubei province in China. Very rapidly, it spread throughout Wuhan and a few other cities in China and also throughout the globe. A new coronavirus-related infectious disease, defined as coronavirus disease 2019 (COVID-19) by the World Health Organization (WHO), is an infectious disease caused by the SARS-CoV-2 virus, which is an RNA virus from the β-coronavirdea family that causes severe pulmonary distress in the infected individuals, and as of 8 May 2022, more than 514 million cases and more than six million deaths have been reported worldwide due to this disease [1,2]. In Turkey, the first case of COVID-19 was seen in March 2020. Soon, the hospitals were overwhelmed by the COVID-19 cases and. for a long time, there was no effective treatment. and no vaccine had been developed. Until now, more than 16 million people have been infected, and there are 100,400 confirmed deaths due to COVID-19 (https://covid19.saglik.gov.tr/) (Accessed on 26 August 2022). Currently, in our country, there are three approved vaccines, two of which are inactivated vaccines, and the remaining is an mRNA vaccine. The Ministry of Health Regulations dictates that individuals should have three doses of vaccines to be considered fully vaccinated. Currently, in Turkey, there are 28,133,966 people who have been vaccinated, which makes up 33.22% of the Turkish population. COVID-19 was declared a pandemic at the end of January 2020, and immediately after, vaccine studies started rapidly all over the world, and currently, there are three COVID-19 vaccines approved by the Food and Drug Administration (FDA). They are all authorized for emergency use [3]. Together with the introduction of COVID-19 vaccines, there was a significant decrease in the number of symptomatic and severe cases as well as COVID-19-related mortality. Despite a few studies reporting adverse effects, the vaccines have been confirmed to be effective and safe in this process [4].

Until the beginning of 2021, there were no effective vaccines. In 2021 the vaccination campaign was initiated. Soon after the vaccination campaign, the disease character changed from severe pneumonia to a mild upper respiratory tract infection. Together with the development came another problem which was vaccine hesitation. WHO has identified vaccine hesitancy as one of the top 10 barriers to global health [5]. Vaccine hesitancy is defined as “delayed acceptance or rejection of a vaccine despite the availability of vaccination services” [6]. Vaccine-hesitant individuals have been defined as a heterogeneous group with varying degrees of uncertainty regarding specific vaccines or vaccination in general [7]. Current vaccines play a key role in keeping the transmission of infectious diseases at a minimum. For this reason, it is very important to vaccinate society and the healthcare professionals who are at the forefront of the struggle within society [8]. The increase in vaccine hesitancy in society is also seen among healthcare professionals. In a systematic review of studies conducted in many countries, it was stated that 22.51% of 76,471 healthcare professionals in total had hesitations about the COVID-19 vaccine [9]. In a study conducted with healthcare professionals in a university hospital in Turkey, it was shown that 82.4% (*n* = 566) of the cleaning personnel and caregivers had been vaccinated, while 17.6% (*n* = 121) did not [10]. It has been observed that healthcare professionals who were quarantined after the severe acute respiratory syndrome (SARS) epidemic in 2003 and working in high-risk areas, such as wards where SARS patients were treated, had anxiety and related symptoms [11]. When the recent studies are examined, it has been shown that healthcare professionals who work in the hospital and have direct contact with COVID-19 patient experience high anxiety and depression [12,13]. During the pandemic, we have observed that the fear of contracting the disease and unrealistic news regarding vaccines have caused significant anxiety and depression among health care professionals. The need for objective validation of these subjective observations has formed the foundations of the present study. The aim of this study was to examine the vaccine hesitancy, anxiety levels, and related factors of the caregivers and cleaning staff working in our hospital during the COVID-19 process.

## 2. Materials and Methods

### 2.1. Type, Place, and Time of Research

This is a descriptive cross-sectional study based on a questionnaire. This research was carried out by using the face-to-face interview technique with the caregivers and cleaning staff working at Inonu University Turgut Ozal Medical Centre between October 2021 and November 2021. All participants gave their informed consent for inclusion before they participated in the study. The study was conducted according to the guidelines of the Declaration of Helsinki. Ethics Committee approval was obtained from the Inonu University Health Sciences Non-Interventional Clinical Research Ethics Committee (2021/2537). STROBE (Strengthening the reporting of observational studies in epidemiology) guideline was utilized to assess the likelihood of bias and overall quality of this study [14].

### 2.2. Population and Sample of Research

Approximately 550 caregivers and cleaning staff actively working in the hospital during the above-mentioned study period were determined as the target population of this study. A priori power analysis suggested that the minimum sample size required to detect a significant difference was 413, considering type I error (alpha) of 0.05, power (1-beta) of 0.9, and effect size of 0.16 for vaccine hesitancy scale and two-sided alternative hypothesis (H1) [15]. Considering the data loss, a total of 475 cleaning personnel and caregivers were interviewed face-to-face, and 460 personnel who answered all questions were included in this study. Primary outcome measures were vaccine hesitancy scale (VHS) and coronavirus anxiety scale (CAS) scores.

### 2.3. Scales Used in the Study

#### 2.3.1. Demographic and Social Characteristics Form

The questionnaire used in this study consists of 28 questions and two scales. The questions querying the socio-demographic characteristics of the study can be briefly defined as follows: age, gender, marital status, education level, smoking, working unit (service, intensive care, emergency room, operating room, outpatient clinics), presence of chronic illness, presence of psychological illness requiring drug use (anxiety, stress, depression), working status in COVID-19 clinics during the pandemic process. It also consists of various questions about the presence of COVID-19 disease, COVID-19 vaccination status (Sinovac, Biontec, both, none), vaccine dose (one, two, three, four doses), general vaccines, and the presence of hesitancy regarding the COVID-19 vaccine and about the COVID-19 vaccine.

#### 2.3.2. Coronavirus Anxiety Scale-Short Form (CAS-SF)

CAS-SF, which aims to determine the severity of anxiety caused by the COVID-19 pandemic in society, was first defined by Lee in 2020 [16]. Cronbach Alpha reliability and internal consistency coefficient of the original study were calculated as 0.93. The validity and reliability tests of the Turkish version of this scale were performed by Biçer et al. in 2020 [17]. The Cronbach’s alpha reliability and internal consistency coefficient of this scale, which was adapted into Turkish, is calculated as 0.832. In CAS scale consisting of five-point Likert-type questions, the scores are ranked as: not at all (0 point), rare, less than a day or two (1 point), several days (2 point), more than seven days (3 point) and nearly every day over the last two weeks (4 point). Lee et al. [16] calculated an optimal cut-off point for anxiety (≥9 points) using ROC curve analysis and calculated the sensitivity and specificity values of this cut-off point as 90% and 85% (AUC: 0.94, *p* < 0.001), respectively. In this scale, where the lowest 0 points and the highest 20 points can be obtained, a score of 9 and above is considered as presence of coronavirus anxiety.

#### 2.3.3. Vaccine Hesitancy Scale (VHS) adapted to Pandemic

The VHS was developed by Larson et al. [18] in 2015 to measure the level of vaccine hesitancy in individuals and possible reasons for it. The Turkish version of this scale was made by Çapar and Çinar [19] in 2021. In the Turkish version, the authors stated that they modified the scale for the pandemic and determined the name of this new version to be “Vaccine Hesitancy Scale in Pandemics”. The answers given to the VHS scale, which consists of five Likert-type questions, are listed as strongly disagree (1 point), disagree (2 points), neither agree nor disagree (3 points), agree (4 points), and strongly agree (5 points). The PVHS scale consists of 10 items and two sub-dimensions. The first sub-dimension is called “lack of confidence”, and all eight items (M1–8) in this sub-dimension are reverse coded. High scores obtained from the lack of confidence sub-dimension indicate that the mistrust towards the vaccine increases in pandemics. The second sub-dimension is called ‘‘risk’’, and the above-mentioned order is used in coding the two items (M9–10) in this sub-dimension. High scores from the risk sub-dimension indicate that the risk of vaccination is high in pandemics. Therefore, when both sub-dimensions are evaluated together, high scores from the PVHS scale show that vaccine hesitancy is high in pandemics. The Cronbach alpha reliability and internal consistency coefficient of this scale, which was adapted into Turkish, is calculated as 0.901.

### 2.4. Statistical Analysis

For statistical analysis, version 25.0 of the SPSS software program was used (Statistical Package for the Social Sciences, Inc., Chicago, IL, USA). The Shapiro–Wilk test of normality was used to show whether the quantitative variables had a normal distribution. Since it was seen that some of the continuous variables did not have a normal distribution, the results were given as median, minimum-maximum, or interquartile range (IQR). Qualitative variables were given as numbers and percentages. The Chi-Square Test was used to compare categorical variables. The non-parametric Mann–Whitney U test was used to compare two independent groups, while the Kruskal–Wallis H test was used to compare three or more independent groups. Bonferroni-corrected Mann–Whitney U test was used for multiple comparisons after the Kruskal–Wallis H test. *p* < 0.05 was considered a statistically significant level.

## 3. Results

### 3.1. Socio-Demographic Characteristics

A total of 460 participants, 322 (70%) male, and 138 (30%) female, aged between 19 and 59 years (median: 30), were included in this study. Fifty-six-point one percent of the participants were married (*n* = 258), 53.9% were high school graduates (*n* = 248) and 70.2% (*n* = 113) had at least one child. Fifty-two-point four percent of the participants stated that they worked in the ward (*n* = 241) and 18.5% in the intensive care unit (*n* = 85). Forty-five-point four percent of the participants stated that they smoke (*n* = 209), and 12.2% stated that they have a chronic disease (*n* = 56). A total of 5.2% of the participants (*n* = 24) stated that they had a disorder that required medication, such as anxiety, stress, or depression (Table 1).

### 3.2. Perceptions of COVID-19 and Vaccines of the Participants

Thirty-six-point-one percent of the participants stated that they had previously contracted COVID-19 (*n* = 166), and 87% (*n* = 401) of all participants indicated that they had the COVID-19 vaccine. Eighty-seven-point-five percent (*n* = 351) of those vaccinated against COVID-19 stated that they had at least two vaccine doses. The rate of participants who were hesitant about all vaccines was 10.9% (*n* = 50), while the rate of hesitation against the COVID-19 vaccine was 42.2% (*n* = 194). The rate of participants who think that COVID-19 vaccines are protective is 44.6% (*n* = 205). The most important issues that the participants are most worried about during the COVID-19 period are their parent contracting COVID-19 in 62.4% (*n* = 287), lack of clear knowledge regarding COVID-19 in 42% (*n* = 193), and own risk of contracting COVID-19 in 36.1% (*n* = 166), respectively (Table 2).

### 3.3. Anxiety and Vaccine Hesitancy Levels of the Participants

The scores obtained from the VHS and CAS scales were calculated as median (IQR; 95% CI). Accordingly, the scores obtained from the VHS-total, VHS- lack confidence, and VHS- risk scales were found to be 35 (11; 34–37), 29 (12; 28–30), and six (3; 6–7), respectively. The score obtained from the CAS score was calculated as five (6; 5–6). When the CAS score was categorized, 19.6% (*n* = 90) of the participants were found to have COVID-19 anxiety (Table 1).

Considering the scores obtained from the CAS scale (cut-off = 9 points), the participants were divided into two groups; those with (*n* = 90) and without (*n* = 370) anxiety. Accordingly, there was no statistically significant difference between the groups in terms of age groups, education level, and the idea of whether the COVID-19 vaccine should be compulsory by law. The COVID-19 anxiety rate was 23.9% in men and 9.4% in women, and there was a statistically significant difference between the groups (*p* < 0.001). Anxiety rates of the personnel working in the intensive care unit and operating room were significantly higher than those working in the polyclinic and emergency services (*p* = 0.002). Anxiety was higher in individuals who did not have a COVID-19 vaccine (*p* = 0.001). A higher rate of COVID-19 anxiety was found in those who had previous psychological disorders requiring medication compared to those who did not (*p* = 0.023) (Table 3).

### 3.4. Evaluation of Relationship between VHS Scores and Various Variables

When the VHS scores of the participants in the research group were compared according to various variables, no significant difference was found between the median VHS-total score in terms of age group, gender, educational status, and contracting COVID-19. The median of the VHS-total score of individuals who are hesitant about the COVID-19 vaccine (29) was significantly higher than the median of individuals who are not (23); the median of the VHS-total score (29) of the individuals who did not receive the COVID-19 vaccine was significantly higher than the median (26) of the individuals who did (*p* < 0.05). The median VHS-total score of those who think that the COVID-19 vaccine is protective was detected to be significantly lower than those who do not believe its effectiveness or who have unclear thoughts, and the median VHS-total score of those who believe that the COVID-19 vaccine should be mandatory by law was detected to be significantly lower than those who do believe that it should be compulsory and who have not decided (*p* < 0.001) (Table 4).

VHS-lack of confidence sub-dimension scores were examined according to various variables. Among the participants in the study, those who did not have the COVID-19 vaccine, who had hesitations about the COVID-19 vaccine, who did not think that the COVID-19 vaccine was protective, who did not believe that the COVID-19 vaccine should be compulsory by law, had a significantly higher VHS-lack of confidence score than the other groups (*p* < 0.05) (Table 4). VHS-risk sub-dimension scores were examined according to some variables. Among the participants in the study group, those who did not have the COVID-19 vaccine, who had hesitations about the COVID-19 vaccine, who did not think that the COVID-19 vaccine was protective, who did not think that the COVID-19 vaccine should be compulsory by law, had a significantly higher VHS-risk score than the other groups (*p* < 0.05) (Table 4).

## 4. Discussion

In the present study, COVID-19 vaccination status, COVID-19 anxiety levels, COVID-19 vaccine hesitancy levels of caregivers and cleaning staff working in the health field during the pandemic period, and affecting factors were examined. In a study examining COVID-19 vaccine hesitancy in healthcare workers, the vaccine hesitancy rate was 42.9% (*n* = 75) among the cleaning personnel. Similarly, in the same study, they were asked about their vaccination status against influenza, and 50.8% (*n* = 91) of this group stated that they had never been vaccinated against influenza in their lifetime [20]. Similarly, nearly half of our study group indicated they were hesitant about the COVID-19 vaccine. Despite this, the rate of individuals who had the COVID-19 vaccine was 87.2% (*n* = 401), and 43.9% (*n* = 176) stated that they had three doses of the vaccine. In a comprehensive study conducted with healthcare workers in Canada, 73.7% (*n* = 638) of nurses and caregivers stated that they had the COVID-19 vaccine. In a cross-sectional study conducted in the USA, 50% (*n* = 5440) of healthcare workers indicated they were hesitant about COVID-19 vaccines [21,22]. The hesitation rate of our research group against all vaccines (10.9%) was lower than the hesitation rate for the COVID-19 vaccine (42.2%). The rate of hesitation against the COVID-19 vaccine in the research group may have been higher than the rate of hesitation against all vaccines because this vaccine is new, and there is a wide variety of false information about the COVID-19 vaccine.

The issues that the healthcare personnel participating in the study were most worried about during the COVID-19 period were expressed as their parents’ contracting COVID-19, the excessive unknowns regarding COVID-19, and their own risk of contracting COVID-19. The vaccination rate in the study group may be high (87.2%) due to the serious concern of COVID-19 transmission to their parents. A study examining the reasons for vaccination among health workers showed that the fear of infecting their families, themselves, and other individuals in the community was the main driving force [22]. Within the research group, the rate of participants who think that COVID-19 vaccines are protective is 44.6% (*n* = 205). Among the personnel participating in the study, individuals who did not receive the COVID-19 vaccine, did not think that it was protective, and did not think that it should be compulsory by law, were found to have higher hesitations about vaccination. A study examining the factors affecting the approval of the COVID-19 vaccine in healthcare personnel showed that the vaccines produced during the epidemic could not be expected to have a safety guarantee due to the rapid process of production. Secondly, lower vaccine acceptance was due to distrust of the Ministry of Health [23]. Similarly, in our study group, those who did not have the COVID-19 vaccine, who had hesitations about the COVID-19 vaccine, who did not think that the COVID-19 vaccine was protective, and who did not believe that the COVID-19 vaccine should be mandatory by law, were found to have higher levels of lack of confidence to the vaccine. The fact that the vaccination program started first among the healthcare professionals during the pandemic and that the process is dynamic and variable, and that the lack of information may have caused a lack of trust in the caregivers and cleaning personnel. However, our results show the need for postgraduate education for healthcare professionals. Furthermore, it would be very useful to share the data regarding the results of vaccination with different types of vaccines. There is also a major impact of the results of educating the health care personnel because they have a greater influence on the population.

When the VHS-total scores of the staff in the research group were compared according to various variables, no significant difference was found between the median VHS total score according to age group, gender, educational status, and the status of contracting COVID-19. Although some studies have found a significant difference between vaccine hesitancy according to age group and gender [21,22,24], no difference has been found in various other studies [25,26]. Vaccine hesitancy is not only affected by socio-demographic variables. Different results may be obtained in accord with a multi-dimensional situation that is examined.

In a systematic review examining the anxiety levels of healthcare workers during the pandemic, anxiety rates were found to be 10–44% [27]. In our study, COVID-19 anxiety was detected in 19.6% (*n* = 90) of the personnel. The reason for the different anxiety rates obtained in the literature and our study may be related to different scales that are used. Furthermore, some of the studies were conducted during the beginning of the pandemic, when there were a lot of unknowns that caused significant anxiety. Furthermore, there were differences in the professions and departments of the health personnel as well as inter-individual differences. In our study, the rate of COVID-19 anxiety was found to be higher in men than in women. In similar other studies examining anxiety, depression, and insomnia in healthcare workers, the anxiety rate was higher in women [28,29]. Since most of our study group consisted of men, it may have affected the results.

In a study examining the psychological state of healthcare workers, anxiety was found to be 2.06 times higher in those who are in contact with COVID-19 patients in the hospital compared to those who did not [29]. Similarly, our study group found the anxiety rates of the personnel working in the intensive care unit and operating room (where contact with infected patients was more frequent) were higher. During the pandemic, hospital cleaning personnel and caregivers had more anxiety when compared to other people because they care for patients in intensive care units. Furthermore, these patients have severe COVID-19, which causes significant anxiety in healthcare professionals. Furthermore, the rate of contact with devices used in operating rooms is higher.

Among the caregivers and cleaning staff in our study groups, COVID-19 anxiety was higher in those who did not receive the COVID-19 vaccine. Caregivers and cleaning staff who did not receive vaccinations may have been more anxious because they knew that the risk of contracting COVID-19 was higher. In addition, among the caregivers and cleaning staff participating in the study, COVID-19 anxiety was found to be higher in those who had previous conditions such as anxiety, stress, or depression that required medication. Similarly, in a study examining the depression, anxiety, and stress states of physicians and related factors during the pandemic, it was shown that individuals with lifetime psychiatric disorders are at higher risk for depression, anxiety, and stress during this period [30]. It may be important because employees with a history of psychiatric illness are more prone to conditions such as anxiety, depression, and stress during the pandemic.

A novel review has reported that future research should focus on effective strategies for not just boosting vaccination rates but also altering underlying attitudes that contribute to vaccine reluctance [31]. In line with this recommendation, our study reports the main findings of attitudes by evaluating the hesitancy of the caregivers and cleaning staff against the COVID-19 vaccine in terms of anxiety, depression, and stress

The present study has some limitations. This questionnaire-based cross-sectional study represents 83.6% of the cleaning personnel in a tertiary health care facility which is enough for representation. However, the results of the present study need validation by other studies before the results can be extrapolated to cleaning personnel in general. This is a very important limitation of the present study. However, multicentric studies during the pandemic are very difficult. Another limitation of our stud is the male predominance of the cleaning personnel. Seventy percent of the personnel were male in our study. However, our results are consistent considering the male predominance in our study.

## 5. Conclusion and Recommendations

The rate of vaccine hesitancy of the caregivers and cleaning personnel participating in the study was high. The hesitation rate against COVID-19 vaccines was almost four times higher than that of the hesitation against childhood vaccines. The most important reasons for this situation are: (i) the rapidly spreading anti-vaccine rhetoric in mass media and social media and (ii) the fact that the COVID-19 vaccine was produced much faster than standard vaccine production procedures. Before the vaccination program started for the general population, the vaccination program was started for the healthcare professionals, and even the phase II and III trials were started for healthcare personnel. The education of healthcare professionals was neglected during this phase because the world needed the vaccine very rapidly. This created great controversy in society. Therefore, we suggest an educational program for all caregivers and cleaning personnel using in-house training or seminars in a way that can be understood.

COVID-19 anxiety was detected in approximately one-fifth of the participants, and those with a condition such as anxiety, stress, or depression that required medication were found to have higher anxiety about contracting COVID-19. Intermittent examinations should be carried out within the framework of occupational health and safety to detect anxiety, depression, and related disorders that may arise during the pandemic, especially in individuals with risk factors, among individuals involved in healthcare, and to provide psychological support to eliminate this situation.

## Figures and Tables

**Table 1 vaccines-10-01426-t001:** Distribution of the participants’ socio-demographic characteristics.

Variables	*n*	%
Gender		
Female	138	30.0
Male	322	70.0
Marital Status		
Married	258	56.1
Single	202	43.9
Educational status		
Secondary school	71	15.8
High school	248	53.9
Associate graduate	80	17.4
Bachelor or postgraduate	61	13.2
Have you child?		
Yes	113	70.2
No	48	29.8
Working unit		
Service (wards)	241	52.4
Intensive care	85	18.5
Emergency unit	44	9.6
Operating room	15	3.3
Outpatient clinic	75	16.3
Smoking		
Yes	209	45.4
No	251	54.6
Do you have a chronic disease?		
Yes	56	12.2
No	404	87.8
Have you a psychological disease (anxiety or depression)		
Yes	24	5.2
No	436	94.8
Age (years)		
Median (IQR)	30 (14)	
95% CI	29–32	
CAS Score		
Median (IQR)	5 (6)	
95% CI	5–6	
VHS Score (Total)		
Median (IQR)	35 (11)	
95% CI	34–37	
VHS Score (Risk)		
Median (IQR)	6 (3)	
95% CI	6–7	
VHS Score (Lack of Confidence)		
Median (IQR)	29 (12)	
95% CI	28–30	

**Table 2 vaccines-10-01426-t002:** Distribution of various variables of participants related to COVID-19 and Vaccination.

Variables	*n*	%
Exposure to the COVID-19?		
Yes	166	36.1
No	294	63.9
Vaccination against COVID-19?		
Yes	401	87.2
No	59	12.8
Number of COVID-19 vaccines		
1 dose	38	9.5
2 doses	175	43.6
3 doses	176	43.9
4 doses	12	2.6
Hesitancy against childhood vaccine		
Yes	50	10.9
No	410	89.1
Hesitancy against COVID-19 vaccine		
Yes	194	42.2
No	266	57.8
Do you think the COVID-19 vaccine is protective?		
Yes	205	44.6
No	109	23.7
No idea	146	31.7
Should the COVID-19 vaccine be made mandatory by law?		
Yes	163	35.4
No	195	42.4
No idea	102	22.2
Which of the following worries you the most, during the COVID-19		
My parents’ exposure to COVID-19	287	62.4
Uncertainties about COVID-19	193	42.0
Individual exposure to COVID-19	166	36.1
Working in the COVID-19 service	71	15.4
Working in the COVID-19 intensive care unit	80	17.4

**Table 3 vaccines-10-01426-t003:** Comparison of COVID-19 Anxiety Status of the Personnel Participating in the Study According to Various Variables.

Variables	COVID-19 Anxiety Status				*p*
	Absence		Presence		
	*n*	%	*n*	%	
Age Groups (years)					0.995
≤20	29	82.9	6	17.1	
21–30	146	81.6	33	18.4	
31–40	111	81.6	25	18.4	
≥41	62	82.7	13	17.3	
Gender					<0.001
Female	125	90.6	13	9.4	
Male	245	76.1	77	23.9	
Educational status					0.397
Secondary school	56	78.9	15	21.1	
High school	196	79.0	52	21.0	
Associate graduate	64	80.0	16	20.0	
Bachelor or postgraduate	54	88.5	7	11.5	
Working unit					0.002
Service (wards)	205	85.1	36	14.9	
Intensive care	56	65.9	29	34.1	
Emergency unit	37	84.1	7	15.9	
Operating room	10	66.7	5	33.3	
Outpatient clinic	62	82.7	13	17.3	
Vaccination against COVID-19?					0.001
Yes	332	82.8	69	17.2	
No	38	64.4	21	35.6	
Have you a psychological disease (anxiety or depression)					0.023
Yes	15	62.5	9	37.5	
No	355	81.4	81	18.6	
Should the COVID-19 vaccine be made mandatory by law?					0.891
Yes	133	81.6	30	18.4	
No	156	80.0	39	20.0	
No idea	81	79.4	21	20.6	

**Table 4 vaccines-10-01426-t004:** Comparison of Vaccine Hesitation Scale Scores and Sub-Dimensions in Pandemics According to Various Variables of Personnel in the Study.

Variables	VHS Risk			VHS Lack of Confidence			VHS Total		
	Median (IQR) (IQR) Score	95% CI	*p*	Median (IQR)	95% CI	*p*	Median (IQR) (IQR) Score	95% CI	*p*
Age Groups (years)			0.757			0.486			0.425
≤20	6 (3)	6–8		18 (12)	15–21		24 (11)	20–28	
21–30	6 (3)	6–7		20 (11)	19–23		27 (12)	26–30	
31–40	6 (3)	6–7		18 (11)	17–22		26 (11)	25–29	
≥41	6 (2)	6–7		20 (14)	18–24		26 (12)	24–30	
Gender			0.351			0.623			0.875
Female	6 (3)	6–8		20 (10)	20–22		26 (10)	26–29	
Male	6 (2)	6–7		19 (13)	18–22		26 (13)	25–28	
Educational Status			0.093			0.398			0.507
Secondary school	6 (3)	6–7		20 (12)	17–24		27 (12)	24–30	
High school	6 (3)	6–7		20 (12)	19–22		27 (12)	25–28	
Associate graduate	6 (2)	6–7		19 (12)	17–21		25 (11)	23–27	
Bachelor or postgraduate	6 (2)	6–8		17 (13)	14–20		24 (13)	20–27	
Exposure to the COVID-19?			0.038						0.994
Yes	6 (3)	6–7		19 (12)	18–21	0.649	26 (12)	25–29	
No	6 (2)	6–7		20 (12)	18–21		26 (13)	26–28	
Vaccination against COVID-19?			0.418			0.042			0.047
Yes	6 (3)	6–7		19 (12)	18–20		26 (11)	26–28	
No	6 (3)	6–8		22 (17)	20–24		29 (16)	24–31	
Hesitancy against COVID-19 Vaccine			0.011			<0.001			<0.001
Yes	6 (3)	6–7		22 (9)	21–24		29 (9)	28–30	
No	6 (2)	6–7		17 (13)	16–18		23 (12)	22–25	
Do you think the COVID-19 vaccine is protective?			0.088			<0.001			<0.001
Yes	6 (3)	6–7		16 (10)	15–18		21 (8)	20–23	
No	6 (2)	6–8		24 (12)	23–27		30 (10)	29–33	
No idea	6 (2)	6–7		23 (8)	21–24		29 (8)	27–30	
Should the COVID-19 vaccine be made mandatory by law?			0.035			<0.001			<0.001
Yes	6 (3)	6–7		16 (12)	15–18		21 (12)	20–23	
No	6 (3)	6–7		23 (10)	23–24		29 (11)	28–30	
No idea	6 (2)	6–7		20 (9)	19–24		26 (10)	26–30	

## Data Availability

The datasets analyzed during the current study are available from the corresponding author on reasonable request.

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
