# Peer review of "Evaluation of Vaccine Hesitancy and Anxiety Levels among Hospital Cleaning Staff and Caregivers during COVID-19 Pandemic"

_vaccines, 2022, doi:10.3390/vaccines10091426_

Round 1

Reviewer 1 Report

In the present study, Akbulut et al. evaluate vaccine hesitancy and anxiety levels in cleaning staff and caregivers, with a representative sample (n=460). Overall, this study supports that these workers present high rates of vaccine hesitancy (app. 45%), and high anxiety levels (app. 1 in 5), which is consistent with previous works evaluating the noteworthy impact of COVID-19 pandemic. 

Despite in general terms the work is properly driven, i have some recommendations for authors: 

1) Although there are no extensive English errors or mistakes, i feel that some parts can be rewritten in order to improve the readibility of the work

2) In material and methods it could be possible that in line 86 is there some data missing? And in line 122 "Bu ölçeÄŸin" should be removed

3) In the section of results, i feel that it would be clearer and better explained if divided in different subsections i.e. 3.1 Sociodemographic characteristics of the participants; 

3.2 Perceptions of COVID-19 and  vaccines of the individuals included

3.3 Anxiety levels in cleaning staff and caregiver workers.....

4) I think that the low proportion of women in comparison to men regarding anxiety in this study could be a relevant limitation, as compelling evidence supports that anxiety is notably higly in women than in men. If possible, i would strongly recommend the authors to include more women in their study, or take it into consideration for future works, specially regarding this kind of disorders. 

5) Finally, i would recommend to add a section or a parragraph of strengths and limitations of the study, in order to put in context the results obtained. 

Author Response

Reviewer 1

In the present study, Akbulut et al. evaluate vaccine hesitancy and anxiety levels in

cleaning staff and caregivers, with a representative sample (n=460).

Overall, this study supports that these workers present high rates of vaccine hesitancy  (app. 45%), and high anxiety levels (app. 1 in 5), which is consistent with previous works evaluating the noteworthy impact of COVID-19 pandemic.

Despite in general terms the work is properly driven, i have some recommendations for authors:

  • Although there are no extensive English errors or mistakes, i feel that some parts can be rewritten in order to improve the readibility of the work
  • In material and methods it could be possible that in line 86 is there some data missing? And in line 122 "Bu ölçeÄŸin" should be removed
  • In the section of results, i feel that it would be clearer and better explained if divided in different subsections i.e.
  • Sociodemographic characteristics of the participants;
  • Perceptions of COVID-19 and vaccines of the individuals included
  • Anxiety levels in cleaning staff and caregiver workers.....
  • I think that the low proportion of women in comparison to men regarding anxiety in this study could be a relevant limitation, as compelling evidence supports that anxiety is notably higly in women than in men. If possible, i would strongly recommend the authors to include more women in their study, or take it into consideration for future works, specially regarding this kind of disorders.
  • Finally, i would recommend to add a section or a parragraph of strengths and limitations of the study, in order to put in context the results obtained.

Response to Reviewer 1 Comments

Thank you very much for your attention and suggestions. The English language of the article was revised in line with your suggestions.

The phrase "Bu ölçeÄŸin" on line 122 has been deleted. Missing data on line 86 was found (550 people) and added to the article text.

In line with your suggestions, four sub-headings have been added to the results section.

The fact that a significant part of the participants included in the study was male is one of the limiting factors in the study. However, when we look at the gender of the personnel in the center where we work, it will be seen that most of the employees are male. We think that this difference between genders in this study should be considered in prospective studies. (Note: taking your suggestions into account, we made an analysis so that the number of women (n=138) and men (n=138) was equal, but even in this case, we found that the anxiety rate was higher in men.

In line with your suggestions, A limitation paragraph has been added to the last part of the article.

Reviewer 2 Report

Important for public health and for future health eduction. I suggest to continue research on this topic.

Author Response

Reviewer 2

  • Important for public health and for future health eduction. I suggest to continue research on this topic.

Response to Reviewer 2 Comments

Thank you very much for your suggestions. I will continue to work on the epidemiology of epidemic diseases.

Reviewer 3 Report

General considerations

The aim of this study was to evaluate the vaccine hesitancy and anxiety levels of hospital cleaning staff and caregivers during the COVID-19 pandemic.

First of all, research is conducted in Turkey and presumably the authors are not native English speakers, the quality of the English language is not sufficient. My first impression is that the paper needs a thorough proofreading and copyediting. I recommend the authors to have their manuscript reviewed by a native English language speaker.

I have few suggestions that I hope the authors find useful.

- Abstract: The abstract does not follow the editorial standard indicated for the journal: “The abstract should follow the style of structured abstracts, but without headings”; therefore, remove the words: Background, Methods, Results and Conclusions. The sample size (N=460) is quite large. The topic is quite interesting.

- Use past tense when discussing the procedure and results as well as other researchers' procedures and results. “The aim of this study was….”

- Keywords: follow the editorial standard indicated for the journal. Too long. Change in “COVID-19; Healthcare professionals; Hesitation; Anxiety”.

A zero should not be inserted before a decimal fraction when the number cannot be greater than 1. For example, p < 0.05 should be written as “p < .05.” Continues in the same way!

Typically, if the exact p value is less than .001, you can merely state p < .001.

Introduction: 

Summarizes recent research related to the topic. The introduction is written short. That is fast to read!

You should give some information about the actual situation in your country.

Use past tense when discussing the procedure and results as well as other researchers' procedures and results.

It is essential to advance the argument/justification about the need for conducting this study.

In the procedure, the authors should explicit that the measures used were created ad hoc, and were not validated instruments; though this information is present in the limitation of the study, it should be included also in the procedures' section.

Author Response

Reviewer 3

  • The aim of this study was to evaluate the vaccine hesitancy and anxiety levels of hospital cleaning staff and caregivers during the COVID-19 pandemic.
  • First of all, research is conducted in Turkey and presumably the authors are not native English speakers, the quality of the English language is not sufficient. My first impression is that the paper needs a thorough proofreading and copyediting. I recommend the authors to have their manuscript reviewed by a native English language speaker.
  • I have few suggestions that I hope the authors find useful.
  • Abstract: The abstract does not follow the editorial standard indicated for the journal: “The abstract should follow the style of structured abstracts, but without headings”; therefore, remove the words: Background, Methods, Results and Conclusions.
  • The sample size (N=460) is quite large. The topic is quite interesting.
  • Use past tense when discussing the procedure and results as well as other researchers' procedures and results. “The aim of this study was….”
  • Keywords: follow the editorial standard indicated for the journal. Too long. Change in “COVID-19; Healthcare professionals; Hesitation; Anxiety”.
  • A zero should not be inserted before a decimal fraction when the number cannot be greater than 1. For example, p < 0.05 should be written as “p < .05.” Continues in the same way!
  • Typically, if the exact p value is less than .001, you can merely state p < .001.
  • Introduction: Summarizes recent research related to the topic. The introduction is written short. That is fast to read!
  • You should give some information about the actual situation in your country.
  • Use past tense when discussing the procedure and results as well as other researchers' procedures and results.
  • It is essential to advance the argument/justification about the need for conducting this study.
  • In the procedure, the authors should explicit that the measures used were created ad hoc, and were not validated instruments; though this information is present in the limitation of the study, it should be included also in the procedures' section.

Response to Reviewer 3 Comments

Thank you very much for your attention and suggestions. The English language of the article was revised in line with your suggestions

Subheadings such as "Background, Methods, Results, Conclusions" in the abstract section have been removed.

Thank you for your positive comments on the sample size and the topic of the study.

In line with your suggestions, past tense was used to present the results.

Thank you very much for your suggestions and support. Keywords have been changed.

P values were revised in line with your suggestions.

We have added the initiation point of the present study in the last sentence of the `Introduction`. Briefly, during the pandemic the anxiety of the healthcare workers have been closely observed and as medical professionals we have felt the same concerns. In addition, during the process of developing the COVID-19 vaccine, a lot of inaacurate information have been released to all media hubs inlcuing the social media which contributed to the increase of anxiety because there was a lot of disinformation regarding the safety of these vaccines. These subjective observations required a confirmation with a study using objective measurement modalities. This provided the initiation point of the present study.

Round 2

Reviewer 3 Report

no comment